# Transiently Transfected Mammalian Cell Cultures: An Adaptable and Effective Platform for Virus-like Particle-Based Vaccines against Foot-and-Mouth Disease Virus

**DOI:** 10.3390/v14050989

**Published:** 2022-05-07

**Authors:** Michael Puckette, Victoria Primavera, Erica Martel, Jose Barrera, William Hurtle, Benjamin Clark, Barbara Kamicker, Mariceny Zurita, David Brake, John Neilan

**Affiliations:** 1Plum Island Animal Disease Center, U.S. Department of Homeland Security Science and Technology Directorate, New York, NY 11944, USA; william.hurtle@st.dhs.gov (W.H.); john.neilan@st.dhs.gov (J.N.); 2Plum Island Animal Disease Center, Leidos, New York, NY 11944, USA; victoria.primavera@tufts.edu (V.P.); jose.barrera@st.dhs.gov (J.B.); barbara.kamicker@st.dhs.gov (B.K.); mariceny.zurita@st.dhs.gov (M.Z.); 3Plum Island Animal Disease Center, SAIC, New York, NY 11944, USA; 4Plum Island Animal Disease Center Research Participation Program, Oak Ridge Institute for Science and Education, New York, NY 11944, USA; martel52@rowan.edu (E.M.); benjamin.a.clark@yale.edu (B.C.); 5BioQuest Associates, LLC, P.O. Box 787, Stowe, VT 05672, USA; david.brake@st.dhs.gov

**Keywords:** virus-like particle, foot-and-mouth disease, foot-and-mouth disease virus, vaccine, swine, cattle, platform, mammalian cell culture, plasmid, DIVA

## Abstract

RNA viruses, such as foot-and-mouth disease virus (FMDV), have error-prone replication resulting in the continuous emergence of new viral strains capable of evading current vaccine coverage. Vaccine formulations must be regularly updated, which is both costly and technically challenging for many vaccine platforms. In this report, we describe a plasmid-based virus-like particle (VLP) production platform utilizing transiently transfected mammalian cell cultures that combines both the rapid response adaptability of nucleic-acid-based vaccines with the ability to produce intact capsid epitopes required for immunity. Formulated vaccines which employed this platform conferred complete protection from clinical foot-and-mouth disease in both swine and cattle. This novel platform can be quickly adapted to new viral strains and serotypes through targeted exchanges of only the FMDV capsid polypeptide nucleic acid sequences, from which processed structural capsid proteins are derived. This platform obviates the need for high biocontainment manufacturing facilities to produce inactivated whole-virus vaccines from infected mammalian cell cultures, which requires upstream expansion and downstream concentration of large quantities of live virulent viruses.

## 1. Introduction

Highly adaptable, rapid-response vaccine platforms are critical for the control of epizootic disease outbreaks and emerging epidemics from serologically diverse viruses. Platform adaptability can be complicated by the need for multiple, properly processed and assembled structural proteins to elicit protection. Such is the case with foot-and-mouth disease virus (FMDV), a member of the Aphthovirus genus of the Picornaviridae family and causative agent of foot-and-mouth disease (FMD), the most contagious viral disease in domestic and wild cloven-hoofed animals. FMD vaccines must present intact capsid structures comprised of 60 protomers, each assembled from four structural proteins, VP1, VP2, VP3, and VP4, derived from a viral-encoded 3C protease that processes the P1 polypeptides [1,2].

FMD is estimated to result in between USD 6.5 and USD 21 billion in annual economic losses in endemic areas [3]. FMD outbreaks in disease-free areas result in the application of international trade restrictions and the culling of susceptible animals. Serologically, FMDV consists of seven serotypes, O, A, Asia, C, SAT1, SAT2, and SAT3, along with multiple strains and topotypes. While serotype O is the most widespread, multiple serotypes are present in FMD endemic areas with virus distribution categorized into different virus pools [4].

Manufacturing of currently licensed FMD vaccines requires the use of high biocontainment production facilities to mitigate accidental release, such as the 2007 FMD outbreak in the United Kingdom [5]. Vaccine production strains must also be constantly evaluated and adapted to circulating FMDV strains to address gaps in vaccine coverage [6]. New FMDV vaccine strain adaptation in cell culture to address these gaps is time-consuming and can be complicated by the loss of protective immunogenicity and poor manufacturing cell line adaptation of specific serotypes, particularly SAT1, SAT2, and SAT3 [7]. Additionally, current inactivated FMD vaccine products require costly downstream purification to remove serologically reactive non-structural viral proteins that can interfere in the current differentiation of infected and vaccinated animal (DIVA) diagnostic assays, further complicating disease surveillance, response, and recovery.

To address these issues, next-generation molecular FMD vaccines, such as the Ad5 vaccine [2,8,9,10] and virus-like particle (VLPs) production platforms [11,12,13,14,15,16] that express only the FMDV proteins required for inducing protective neutralizing antibodies, have been developed. Although these newer FMD vaccine platforms are efficacious in cattle, protecting swine has been more difficult, particularly against the widest circulating FMDV O serotype and topotypes/lineages [17,18]. These platforms can also have additional drawbacks, including the prohibitively high cost of goods and/or the need for capsid modifications to stabilize antigenic structures for expression in non-native cellular environments.

To address the drawbacks of currently inactivated vaccines and replication-deficient adenovirus-based vaccine platforms, we developed a plasmid-based, transient mammalian cell transfection platform to produce VLPs. Mammalian cell culture produced FMD VLPs (mamVLPs) are expressed, processed, and assembled in the virus’s native environment. This allows for the use of nonmodified P1 polypeptide sequences, enabling rapid adaptability of the platform to new topotypes and subtypes. The absence of live FMDV in the manufacturing process enables vaccine production in low biocontainment facilities. Consequently, this expands the number of infectious disease vaccine manufacturing facilities that can be used for production scale-up, should vaccine manufacturing surge capacity be needed.

The assembly of VLPs in transiently transfected cell cultures has been observed previously; however, the yield of these VLPs was limited by the detrimental effect of the FMDV 3C protease [2,19,20,21]. The development of the 3C^(L127P)^ mutant protease, which demonstrates reduced host cell protein processing abilities [22], offers a means to overcome previous limitations in this platform and provide a more efficient means to produce VLPs.

## 2. Materials and Methods

### 2.1. Plasmid Constructions

The pJJPKan plasmid consists of the pJJP backbone plasmid [22,23], with the kanamycin resistance gene replacing the ampicillin resistance gene, as shown in Figure 1. Plasmids expressed the P1-2A variable region of the target FMDV strain and the FMDV 3C^(L127P)^ protease mutant, derived from FMDV Asia1 Lebanon 1989 [24], in a single open reading frame, presented in Figure 1. Incorporation of the 3C^(L127P)^ mutation is critical due to the reduction process of host cellular proteins [22]. Nucleotide sequences for P1-2A of target strains were synthesized by Genscript and inserted into the plasmid using the *BamHI* and *ApaI* restriction sites, as shown in Figure 1. Nucleotide sequences were derived from the following GenBank accessions with silent mutations utilized to remove necessary cut sites: A24 strain Cruzeiro GI: AY593768; Asia 1 isolate As1/Shamir/89 GI: JF739177; C3 strain Indaial GI: AY593806; O1 strain Manisa GI: AY593823; SAT 1 isolate KNP/196/91 GI: MN442538; SAT 2 isolate KNP-19-89 GI: KR108949; and SAT 3 isolate ZIM/05/91/3 GI: DQ009740.

### 2.2. Mammalian Cell Culture Transfections

Transfection grade linear polyethylenimine (PEI) of molecular weight 25,000 (Polysciences, Warrington PA) was prepared from powder, as per the manufacturer’s instructions. The prepared liquid PEI was stored in 1 mL aliquots at −20 °C. For transfection, HEK293T/17 cells (CRL-11268; ATCC, Manassas, VA, USA) were cultured in T-75 flasks to roughly 90% confluence. In a 1.5 mL micro-tube (Eppendorf), 32 μg of plasmid DNA was mixed with 50 μL of OptiMEM (Thermo Fisher Scientific, Waltham, MA, USA) media, while in a separate 1.5 mL Eppendorf tube, 96 μL of PEI reagent was mixed with 50 μL of OptiMEM (Thermo Fisher Scientific, Waltham, MA, USA) media. Tube contents were combined and incubated at room temperature for 5 min before transfer to T-75 flasks. Flasks were incubated for 24 h at 37 °C in a 5% CO_2_ incubator prior to harvest.

### 2.3. Transmission Electron Microscopy

Transmission electron microscopy (TEM) was performed, as previously described [23].

### 2.4. Preparation of mamVLP Vaccine

Media were removed and transfected cells were rinsed with 10 mL of 1× dPBS (Thermo Fisher Scientific, Waltham, MA, USA). Cells were detached from the flask by pipetting with 10 mL of culture media. Detached cells were pelleted at 500× *g* for 10 min at 4 °C. Supernatant was removed, and the cell pellet was resuspended by pipetting with 2 mL of 1× dPBS, followed by centrifugation at 500× *g* for 10 min at 4 °C. Wash buffer was removed from the cell pellet, and the wash repeated with fresh 2 mL of 1× DPBS.

Cell pellets were resuspended in 1 mL of Lysis buffer 9 (LB9) (20 mM Tris-HCl, 200 mM NaCl, 3 mM MgCl_2_, 0.1% Triton-X) and incubated at room temperature for 10 min on a rocking platform followed by centrifugation at 1000× *g* for 10 min at 4 °C to pellet cell debris. Supernatant was applied to a 1,000,000 Da MWCO Vivaspin 20 centrifugal concentrator (Viva products, Littleton, MA, USA) and centrifuged for 5 min at 2000× *g*, and the retained volume was stored as 1 mL aliquots at −70 °C.

This extraction methodology produces vaccine preparate containing proteins from the production cell line in addition to VLPs. A Coomassie-stained SDS-PAGE gel of an O1M vaccine preparate is presented in Appendix A.

### 2.5. Cesium Chloride Gradients

Culture media from FMDV-infected cells or extracted VLPs were layered on 2 mL cesium chloride 2-step discontinuous gradients (1.42 g/cm^3^/1.38 g/cm^3^) prepared in TEN buffer (0.05 M Tris, 0.001 M EDTA, 0.15 M NaCl, pH 7.4). Samples were centrifuged in a SW40Ti rotor at 217,485× *g* for 18 h using an Optima L-80 XP ultracentrifuge (Beckman Coulter, Brea, CA, USA). Individual visible bands were collected and dialyzed against PBS at 4 °C using 10 K MWCO Slide-A-Lyzer dialysis cassettes (Thermo Fisher Scientific, Waltham, MA, USA).

### 2.6. Swine Studies

Prior to conducting swine studies, approval of experimental protocol was obtained from the Plum Island Animal Disease Center Institutional Review Board and the Institutional Animal Care and Use Committee (protocol code 240.01-19-D, approved 18 June 2019). Predominantly male Yorkshire pigs, approximately 2.5 months old, were enrolled. Swine were vaccinated with 1.2 mL, 1.6 mL, or 2.0 mL of vaccine formulated with Montanide ISA 201 VG (Seppic SA, Codex France) and containing approximately 12, 16, and 20 μg of antigen, as determined by ELISA, respectively. Swine receiving 12 and 16 μg dosages received a second dose of vaccine 28 days after the initial vaccination and 7 days prior to challenge.

Vaccine efficacy was evaluated in a direct contact challenge exposure model in which one group of naïve swine (*n* = 5) was temporarily separated and infected by intradermal heel bulb inoculation with 100 pig heel infectious dose 50% (PHID50) of FMDV O1 Manisa, and served as FMDV ‘seeders’. The following day, either 14 days post-vaccination for single dosage or 35 days after the initial vaccination for two dosages, seeders were co-mingled with all animals. A control group (*n* = 5) inoculated with PBS served as both sentinels to verify transmission from infected seeders and as virus amplifiers. Temperatures were taken daily during the duration of the study. Animals were observed daily for 14 days for clinical signs of FMD, and any positive animals were removed from the study.

### 2.7. Cattle Study

Prior to conducting cattle studies, approval of experimental protocol was obtained from the Plum Island Animal Disease Center Institutional Review Board and the Institutional Animal Care and Use Committee (protocol code 266.00-20-D, approved 13 November 2020). Female Holstein cattle of approximately 7 months of age and weighing between 450 and 545 lbs were enrolled. Cattle were vaccinated with 1.6 mL of formulated vaccine containing approximately 16 μg of antigen and Montanide ISA 201 VG (Seppic SA, Codex France), as described above. Cattle were directly challenged at 21 days post-vaccination via the intradermolingual (IDL) route with 1 × 10^4^ bovine tongue infectious dose in a total volume of 0.4 mL distributed across four injection points, 0.1 mL each. Rectal temperatures were taken daily during the duration of the 3-week post-challenge study, and animals were observed 7, 14, and 21 days post-challenge for clinical signs of FMD (oro-nasal and pedal lesions). Serum and plasma samples were collected 0, 7, 14, and 21 days post-vaccination, and 7, 14, and 21 days post-challenge.

### 2.8. Virus-Neutralizing Test Antibody Titer Assay

Virus-neutralizing antibody titers (VNTs) against FMDV O1 Manisa were determined on BHK-21 cells (ATCC CCL-10), as per the World Organization for Animal Health protocols [10,25]. Neutralization titers were expressed as the log10 of the reciprocal of the highest serum dilution resulting in 50% neutralization of the cytopathic effect (Spearman–Kärber method).

### 2.9. Viremia Detection

Plasma samples were evaluated with a real-time reverse transcriptase polymerase chain reaction (rRT-PCR) for FMDV O1 Manisa nucleic acid. RNA was extracted with the Applied Biosystems™ (Thermo Fisher) MagMAX™ Pathogen RNA/DNA Kit, and rRT-PCR was performed on an Applied Biosystems ABI 7500 platform using Applied Biosystems™ (Thermo Fisher) TaqMan™ Fast Virus 1-Step Master Mix with the following probes: (forward) 5′-ACTgggTTTTA CAAACCTgTga-3; (reverse) 5′gCgAgTCCTg CCACggA-3; and 6FAM-TCCTTTg CACgCCgTgggAC-Tamra. The presence of a Ct value < 40 was considered positive, while ≥40 was scored negative.

### 2.10. FMDV Nonstructural Protein ELISA

The FMDV antibody test kit (VMRD), a competitive ELISA, was used to detect FMDV 3B nonstructural protein antibodies on serum samples collected on days 21 dpv/0 dpc, 35 dpv/14 dpc, and 42 dpv/21 dpc. A sample was considered positive if the percent inhibition was ≥40%.

## 3. Results

### 3.1. Production Using Transiently Transfected Mammalian Cell Culture

DNA plasmids encoding the P1 capsid polypeptide and 3C protease in a single open reading frame were transfected into HEK293T/17 mammalian cells utilizing polyethylenimine (PEI), as outlined in Figure 2. P1 polypeptide expression and processing resulted in FMDV mamVLPs assembled into crystalline arrays, [24,25,26], as visualized by TEM, shown in Figure 2. Cells were lysed between 21 and 24 h post-transfection using LB9, comprised of low-cost components. The VLPs were concentrated using a centrifugal concentrator column with a 1000 kDa molecular weight cutoff and stored at −70 °C prior to final formulation with adjuvant and administration to animals.

### 3.2. Vaccination and Challenge of Swine and Cattle

The efficacy of mamVLP vaccines was evaluated by administrating a single vaccine dose to cattle and both single and dual vaccine doses to swine. Following vaccination, animals were infected with FMDV O1 strain Manisa (O1M) via contact exposure in swine and direct inoculation in cattle. Cattle were directly challenged by intradermolingual inoculation of 1 × 10^4^ bovine tongue infectious doses, while swine were challenged using a contact challenge model in which five swine were subject to heel bulb inoculation with 100 PHID50 of virus and allowed to intermix with both control, *n* = 5, and vaccinated individuals. Upon challenge, mock vaccinated control animals for both swine (*n* = 10) and cattle (*n* = 2) displayed full clinical disease. Full protection from clinical FMD and viremia was observed following a single dose of the mamVLP given to cattle or after a dual dose of mamVLP administered to swine, as presented Table 1. Swine inoculated with a single dose of the mamVLP vaccine failed to develop protection against clinical FMD despite seroconversion of all animals after vaccination. VNTs at the time of challenge for 2 of 5 swine were comparable to those in swine protected by the two-dose series, as displayed in Figure 3. In addition to protection from clinical FMD and viremia, 4 of 5 of cattle and 7 of 10 of swine were fully protected from fever, as shown in Table 1. Individual VNT titers and rectal temperatures are available in a Appendix A.

Serum samples from vaccinated cattle were evaluated 0, 14, and 21 days post-challenge for the presence of non-structural protein antibodies by commercial ELISA. None of the cattle were seropositive on the day of challenge (doc), as presented in Table 2, confirming the DIVA compatibility of this platform using licensed diagnostic tests [27,28]. After challenge, only a single calf, D21-06, became seropositive, as Table 2 shows.

### 3.3. Platform Adaptability

To test the pan-serotype versatility of the mamVLP platform and utilized reagents, plasmid constructs were produced, representative of the other six serotypes: A24 strain Cruzeiro (A24), Asia 1 isolate As1/Shamir/89 (Asia1), C_3_ strain Indaial (C_3_I), SAT1 isolate KNP/196/91 (SAT1), SAT2 isolate KNP-19-89 (SAT2), and SAT3 isolate ZIM/05/91/3 (SAT3). Transfected cells were visualized by TEM, and harvested cells were evaluated by gradient sedimentation and Western blot. Three of the seven major serotypes (O, Asia, and SAT2) demonstrated consistent presentation of cellular VLP arrays in transfected cells, as detailed in Figure 4A. Notably, Asia1 constructs consistently produced numerous large intracellular arrays, shown in Figure 2. Cellular arrays were not observed by TEM in A24, C_3_I, SAT1, SAT2, or SAT3 transfected cells. However, analysis by gradient density sedimentation demonstrated distinct banding in the 1.38 g/cm^3^ fraction, consistent with properly intact antigen, as displayed in Figure 4B. The Western blotting of extracted bands confirmed the presence of fully processed VP2 in A24, Asia1, C_3_I, and SAT3 constructs, as Figure 4C shows.

## 4. Discussion

While numerous studies have examined the immunogenicity of FMDV VLPs produced by other means, such as baculovirus or adenovirus systems, in non-target species, [13,14,15,16], few have examined the ability of VLPs to protect target species, both swine and cattle, from clinical disease upon challenge. This report demonstrates full protection in cattle, five of five, from challenge with a O1M mamVLPs using a single shot, presenting an improvement upon reported results using the baculovirus system, which either produced partial protection, two of four, in cattle against A22, [29], or required a two-dose regiment against O1M, [30]. We observed full protection of swine from clinical disease using a two-dose series at two different dosages: 12 μg and 16 μg. This improves on previous reports of protection of two of four swine against A12 using a three-dose series of baculovirus-produced antigen [31]. This enhancement in protection may be due to utilization of the 3C^(L127P)^ mutant protease and corresponding increase in transgene expression and independent of the VLP expression platform [22].

The presence of neutralizing antibodies post-vaccination is often used as a surrogate to screen vaccine efficacy prior to challenge. Despite divergent clinical results between swine receiving two doses of 16 μg, i.e., five of five protected, and those receiving a single 20 μg dose, i.e., zero of five protected, observed VNTs were comparable, as shown in Figure 3. These results suggest that for VLP vaccines in swine, an individual’s neutralizing antibody titers cannot be used as a proxy for clinical protection.

Similarly in cattle, only four of five had VNTs above those of unvaccinated animals at the time of challenge, shown in Figure 3, despite all five cattle being protected from clinical disease, shown in Table 1. The same individual, D21-06, with VNTs equivalent to unvaccinated animals at the time of challenge did became febrile post-challenge, but did not demonstrate lesions or observed viremia, as evident from Table 1. It is possible that viremia was present in this individual but did not align with sample collection time points, and was not significant enough to elicit clinical FMD. Individual D21-06 was also the sole positive result in the 3B ELISA, supporting the conclusion that a low level of viremia was present and demonstrates that usage of DIVA compatible vaccine platforms may allow for usage of the 3B ELISA for screening of individuals with infections not demonstrating clinical lesions.

Plasmid constructs containing the P1-2A sequence of six additional serotypes were produced and utilized for the production of VLP antigen, as presented in Figure 4. While only three serotypes demonstrated observable arrays by TEM, O1M, Asia1, and SAT2, all six additional serotypes did demonstrate appropriately sized antigen by gradient centrifugation, as shown in Figure 4C, four of which, A24, Asia1, C_3_I, and SAT3, showed strong reactivity to the F1412SA monoclonal antibody on Western blot. The lack of reactivity of fully processed VP2 by Western blot in SAT1 and SAT2 constructs was most likely due to mAb epitope specificity, as reported previously [22]. These results demonstrate that the mamVLP platform is highly adaptable to FMDV strains and serotypes, including the three FMDV SAT serotypes often associated with wildlife reservoirs, such as the African buffalo (*Syncerus caffer*).

## 5. Conclusions

The work presented here represents a proof-of-concept testing of FMDV vaccines produced using transiently transfected mammalian cell culture. Vaccines produced in this system were capable of protecting both cattle and swine from clinical disease following challenges with the FMDV strain O1 Manisa. Additional work will be required to establish the minimum protective dosage for both cattle and swine to better determine the economics of the platform.

The mamVLP platform provides unique benefits. Its adaptability has the potential to allow for regional manufacturing of FMD vaccines relevant to specific geographic regions to improve vaccine coverage. This versatile platform can also be used to incorporate other emerging FMDV molecular vaccine technological advancements, such as capsid stabilization mutations [29,32] or affinity tags for purification [33,34]. While these targeted changes may reduce the speed of platform adaptation to any new single strain, they may be useful to produce mamVLP-concentrated antigen stockpiles against established high-threat strains. Most importantly, the production of mamVLPs precludes the accidental release of viable FMDV from vaccine production facilities, allowing for safe vaccine manufacturing in FMD-free areas.

## 6. Patents

Patents covering work included in this manuscript include US 10,858,634 and US 11,191,824.

## Figures and Tables

**Figure 1 viruses-14-00989-f001:**
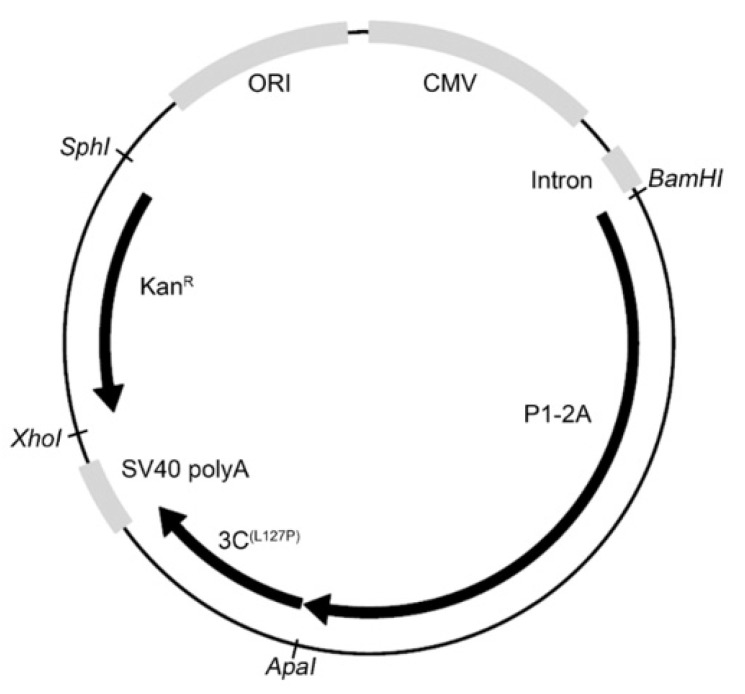
Plasmid map of a pJJPKan VLP expression plasmid containing the FMDV P1-2A and 3C^(L127P)^ sequences in a single open reading frame. Restriction sites *BamHI* and *ApaI* are utilized to insert the P1-2A sequence from any desired serotype or strain into the plasmid.

**Figure 2 viruses-14-00989-f002:**
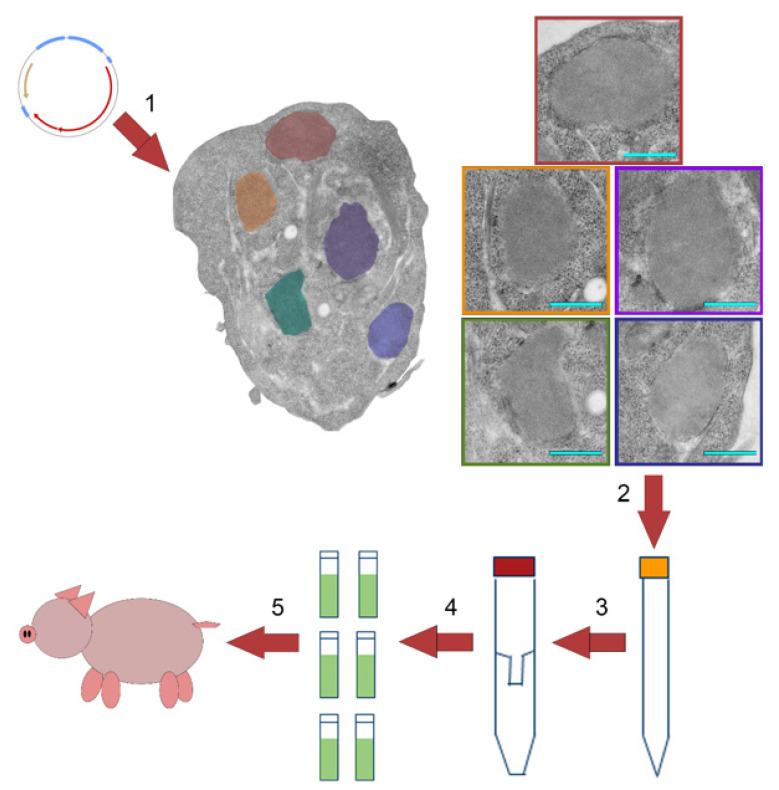
Virus-like particle (VLP) production process using transiently transfected mammalian cell cultures. (1) DNA plasmids encoding the FMDV P1 and 3C protease sequences were transfected into cell cultures using polyethylenimine. Transfected cells were incubated for 24 h to allow expression and processing of the P1 polypeptide resulting in the formation of cellular VLP arrays. Enlargement of individual arrays are presented inside of color-coded boxes; blue bar represents 500 nm. (2) VLPs were harvested by lysis of cells using LB9 and (3) concentrated by a centrifugal concentrator column. (4) The retained high molecular weight fraction was aliquoted into doses and stored frozen until (5) administered to swine or cattle.

**Figure 3 viruses-14-00989-f003:**
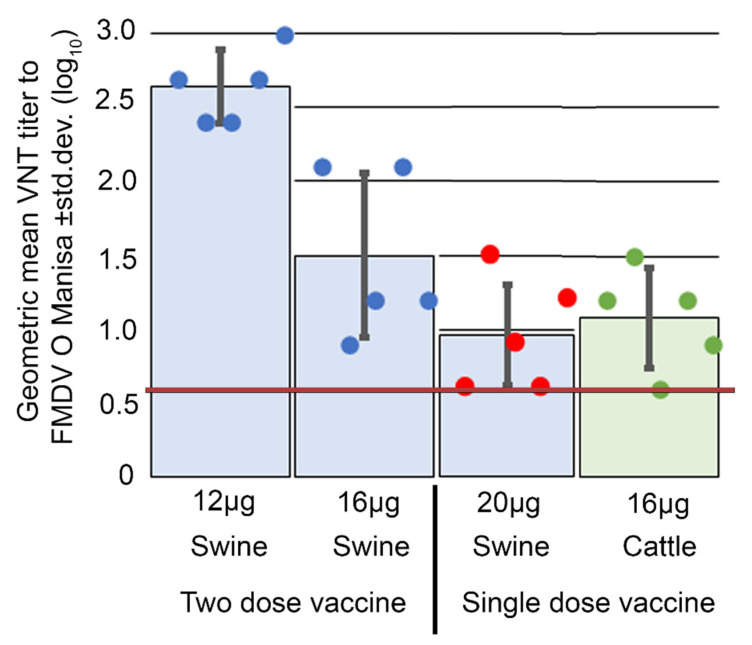
VNTs at the time of challenge for treatment groups and individuals. The mean VNTs for swine treatment groups, 12 μg, 16 μg, and 20 μg, are presented in blue, while the mean VNTs for the 16 μg cattle group are presented in green. Protected swine are represented by blue circles, protected cattle are represented by green circles, and unprotected swine represented are represented by red circles, all grouped by treatment. Baseline VNTs observed in unvaccinated animals, both swine and cattle, are 0.6 and represented by a red line.

**Figure 4 viruses-14-00989-f004:**
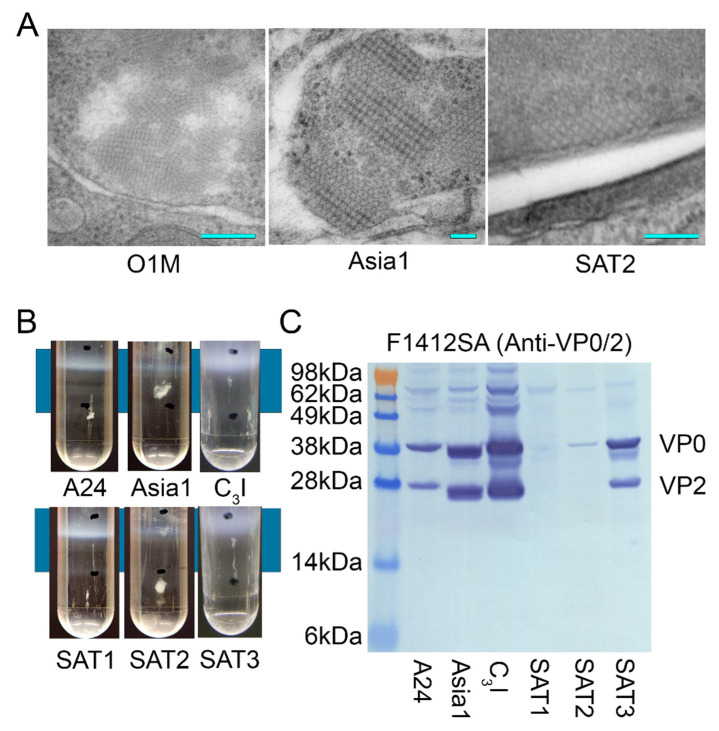
(**A**) Cellular VLP array images captured in cells transfected with constructs expressing FMDV O1M, Asia1, and SAT2 containing constructs. The light blue bar represents 100 nm. (**B**) The visual bands present in the 1.38 g/cm^3^ fraction, localized between tube markings, of cesium chloride gradients demonstrated an antigen of the correct size for VLPs in serotypes A24, Asia1, C_3_I, SAT1, SAT2, and SAT3. (**C**) Western blot of the VLP band from cesium chloride gradients utilizing the F1412SA monoclonal antibody, reactive with VP2 and its progenitor VP0, confirmed the presence of both VP2 and VP0 in VLPs.

**Table 1 viruses-14-00989-t001:** Vaccination parameters and clinical results for cattle and swine. Full protection from clinical FMD was achieved with a single mamVLP dose in cattle and two doses of mamVLPs in swine. Protection from fever is defined as rectal temperatures ≤ 39.6 °C, sampled daily, and protection from viremia is defined as negative RT-PCR (Ct < 40) at sampled time points (3, 7, or 8, and 14 days post-challenge for swine and 7, 14, and 21 days post-challenge for cattle). The geometric mean virus neutralization antibody titers (VNTs) with standard deviation (SD) on the day of challenge (doc) are included. Control animals were not sampled for viremia as all animals displayed clinical disease. * Two swine demonstrated clinical FMD lesions before demonstrating fever and were euthanized.

				Protection From		
Species	Dose (μg)	Regimen	Doc Avg VNT(SD) (Log_10_)	Clinical Disease	Fever	Viremia
Cattle	16	1-dose	1.08 (0.34)	5 of 5	4 of 5	5 of 5
Cattle	0	Control	0.60 (0.00)	0 of 2	0 of 2	N/A
Swine	20	1-dose	0.96 (0.39)	0 of 5	0 of 5	0 of 5
Swine	12	2-dose	2.64 (0.25)	5 of 5	5 of 5	5 of 5
Swine	16	2-dose	1.50 (0.56)	5 of 5	2 of 5	5 of 5
Swine	0	Control	0.60 (0.00)	0 of 10	0 of 8 *	N/A

**Table 2 viruses-14-00989-t002:** Detection of antibodies to FMDV nonstructural proteins utilizing 3B ELISA in post-vaccination/pre-challenge and post-challenge vaccinated cattle. Positive values categorized as inhibition ≥ 40%.

Animal ID	Days Post-Challenge % Inhibition		
	0	14	21
D21-03	7	13	13
D21-05	13	23	14
D21-06	4	72	68
D21-07	13	21	24
D21-10	17	21	14

## Data Availability

Data from animal study experiments can be found in Appendix A.

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
