# Peer review of "Transiently Transfected Mammalian Cell Cultures: An Adaptable and Effective Platform for Virus-like Particle-Based Vaccines against Foot-and-Mouth Disease Virus"

_viruses, 2022, doi:10.3390/v14050989_

Round 1

Reviewer 1 Report

This paper sets out to show VLPs can be quickly and efficiently produced from transient transfection of mammalian cells and used effectively as vaccine candidates in cattle and swine. While I agree results were very encouraging, I would worry about the scalability of this process (acknowledged in lines 306-7), however this is a proof of concept paper, so will treat it as that.

Line 65-7. I wouldn’t agree with the statement that the current VLP production methods using virus vectors carry high costs; they are an efficient way of producing VLPs. As for the stabilising mutations, they are introduced when designing the constructs, incurring no additional costs, plus allow the VLPs extended shelf life. The advantage that this current paper's method has, is that recombinant viruses don't have to be made.

Line 140 details of ELISA?—or refs?

Line 194-5 /202-4 Were samples of VLPs taken to assess how intact the VLPs were before animals were vaccinated?

In section 3.2 (line 205) Please comment on different challenge methods used.

Fig 4  (line 254) Maybe include a Coomassie gel to show all capsid proteins?

Line 272-4 surely the 3C mutation just allowed enhanced VLP expression…not sure why it alone would lead to enhanced protection.

Author Response

Point 1: Line 65-7. I wouldn’t agree with the statement that the current VLP production methods using virus vectors carry high costs; they are an efficient way of producing VLPs. As for the stabilising mutations, they are introduced when designing the constructs, incurring no additional costs, plus allow the VLPs extended shelf life. The advantage that this current paper's method has, is that recombinant viruses don't have to be made.

Response 1: The authors agree that each vaccine platform contains its own benefits and drawbacks and the statement of lines 65-67 is not meant to imply that all platforms contain all potential drawbacks with no benefits.  We have added the word “can” in line 65 to better define that these drawbacks may not be present in all platforms as well as to emphasize that they may only be needed for expression in non-native cellular environments.

Point 2: Line 140 details of ELISA?—or refs?

Response 2: ELISA quantification was performed by an industry partner utilizing proprietary reagents and methodology.

Point 3: Line 194-5 /202-4 Were samples of VLPs taken to assess how intact the VLPs were before animals were vaccinated?

Response 3: Samples from these exact production batches were subject to quantification by ELISA and analysis by western blotting.  During the development of the lysis buffer 9 methodology multiple production batches were tested by cesium chloride gradient to ensure retention of appropriate density antigen following separation by cesium chloride density gradients. 

Not included in this data set is a significant amount of work with pilot studies using a previous extraction methodology that elicited neutralizing epitopes in non-challenged guinea pigs, swine, and cattle as well as preliminary challenge data with swine using the previous extraction methodology.  VLPs in those studies were also analyzed by cesium chloride gradients, such as those in Figure 4B, to determine if antigen was retained in the appropriate fraction.  This provided us with a post-extraction antigen profile to match during development of the lysis buffer 9 methodology and gave us a high confidence of protection from clinical disease going into animal studies containing challenge with FMDV.

Point 4: In section 3.2 (line 205) Please comment on different challenge methods used.

Response 4: Added the following sentence into the results to summarize the challenge methodologies “Cattle were directly challenged by intradermolingual inoculation of 1x104 bovine tongue infectious doses while swine were challenged using a contact challenge model in which five swine were heel bulb inoculation with 100 PHID50 of virus and allowed to intermix with both control, n = 5, and vaccinated individuals.”

Point 5: Fig 4  (line 254) Maybe include a Coomassie gel to show all capsid proteins?

Response 5: Our Coomassie gels demonstrate that while individual VPs are present in the vaccine there is carry over of other proteins as well making it difficult to discern discrete bands associated with individual VPs, gel image included below.  This is the result of utilizing a lysis buffer on transfected cells to extract antigen.  While not producing a highly purified vaccine the current extraction methodology presented here accomplished the task of serving as a proof of concept for the system. Further work with this platform would ideally focus on transition to an industrial process in which additional purification steps could be incorporated.

Point 6: Line 272-4 surely the 3C mutation just allowed enhanced VLP expression…not sure why it alone would lead to enhanced protection.

Response 6: As stated in lines 272- 274, the authors agree that it is unlikely that the 3C mutation creates a more immunogenic or more stable VLP and more likely that enhanced protection is a result of a greater abundance of VLPs in the final product.  It is possible that the system the VLPs are being produced in does play a role.  In the case of references 30, 31, and 32 the VLPs were all produced using baculovirus in insect cell culture.  The data presented here produced VLPs in their native environment, that of the mammalian cell, and it is possible that this has an effect on how epitopes are presented in the vaccinated animal.  This is speculative however and the authors have no data in regards to this.

Reviewer 2 Report

Puckette et al. use transient transfection of mammalian cells to produce FMDV VLPs. They use VLPs to vaccinate cattle and swine and observe protection of the animals against virus challenge.

Major comments:

1) The quality of the VLP preparate used for vaccination must be characterized. A minimal characterization would include the following:

-Protein purity (SDS-PAGE)

-DNA content

-Endotoxin levels

-Particle size distribution (preferably DLS analysis)

Optional: TEM analysis for the purified particles.

2) The results for the challenge study carried out for non-vaccinated animals should be shown with the same analyses as performed for the vaccinated animals (Table 1)

Minor comments:

3) Figure 2: the particle arrays are not visible due to poor resolution of the pdf file

4) Table 1 requires some more explanation: The viremia is studied in 4 different time points. Would "Protection from viremia" mean that elevate temperature wasn't observed in any of the time points? The same question applies to Fever.

5) According to Table 1, one of the animals experienced Fever. Was it the same animal which experienced 3B seropositivity?

6) Terminology: VLP array ... Cellular array. I would suggest to use term "VLP array" or "Cellular VLP array"

7) I wonder if Figure 4C is a western blot analysis or is it rather SDS-PAGE analysis with total protein staining?

I would suggest to present both SDS-PAGE with total protein staining and western blot analysis.

Author Response

Point 1: The quality of the VLP preparate used for vaccination must be characterized. A minimal characterization would include the following:

-Protein purity (SDS-PAGE)

-DNA content

-Endotoxin levels

-Particle size distribution (preferably DLS analysis)

Optional: TEM analysis for the purified particles.

Response 1: This work represents a proof-of-concept analysis of the platform and not a data set to be used for filing with a regulatory body.  As such, characterization of the VLP preparate was more limited.  During the development of the extraction procedure particle density was evaluated by cesium chloride gradient to ensure that particles of the appropriate density for empty capsids were present when using the vaccine preparation methodology.  VLP preparate was also tested in non-challenge studies in Guinea pigs, swine, and cattle to determine safety and ability to elicit neutralizing antibody titers prior to undertaking challenge studies. 

The VLP preparate represents a mixed sample resulting from lysis of the cell, when an SDS-PAGE gel is stained using Coomassie blue, banding at appropriate weights for individual VPs is present, see image included below.  Additional proteins are also present that do not react with FMDV antibodies and most likely represents cellular proteins also extracted from the producing cell line at the same time.  Because of this, the authors chose not to quantify total protein using a Bradford assay and instead quantified the antigen used for vaccination with a proprietary ELISA made available by an industry partner.   Antigen was evaluated by western blotting as well to ensure the presence of FMDV antigen prior to submission of samples to the industry partner for testing.  Endotoxin levels were not evaluated as these vaccines were not produced utilizing bacteria. No negative reactions following vaccination was observed in any of the vaccinated animals, either cattle or swine.

This work was performed on a benchtop scale utilizing T-75 flasks and conversion to either pilot production or manufacturing production scales is likely to result in changes to methodologies utilized, certainly in the scale of cell cultures and possibly extraction methodologies.  The authors agree that any final commercial product would incorporate an enhancement in purification beyond that presented here. 

The authors feel the data presented in this report does successfully demonstrate that antigen produced in transiently transfected mammalian cell culture can be used to elicit protection from clinical FMDV in both swine and cattle.  This demonstration is critical for justification of further investment to scale up the process and begin the process to develop the tools and methodologies needed for commercialization.

Point 2: The results for the challenge study carried out for non-vaccinated animals should be shown with the same analyses as performed for the vaccinated animals (Table 1)

Response 2: Added data from the control animals in Table 1.  Data for VNTs from control animals was also added into the supplemental file containing all animal study data.  The supplemental file also contains temperature data for these control animals.

Point 3: Figure 2: the particle arrays are not visible due to poor resolution of the pdf file

Response 3: A higher resolution Figure 2 has been provided to the editor separately as file size limitations prevented its uploading in the initial submission.

Point 4: Table 1 requires some more explanation: The viremia is studied in 4 different time points. Would "Protection from viremia" mean that elevate temperature wasn't observed in any of the time points? The same question applies to Fever.

Response 4: Yes, the authors interpret protection from fever to be an absence of observed fever at time points samples while protection of viremia is the absence of virus in samples.  Due to the less invasive nature, more time points were taken looking at fever as opposed to viremia.  Added “sampled daily” into the description of Table 1 to better clarify the differences in data points between fever and viremia.

Point 5: According to Table 1, one of the animals experienced Fever. Was it the same animal which experienced 3B seropositivity?

Response 5: Yes, to clarify that the authors have reworded a sentence to state “Individual D21-06 was also the sole positive result in the 3B ELISA, supporting the conclusion that a low level of viremia was present and demonstrates that usage of DIVA compatible vaccine platforms may allow for usage of the 3B ELISA for screening of individuals with infections not demonstrating clinical lesions.”

Point 6: Terminology: VLP array ... Cellular array. I would suggest to use term "VLP array" or "Cellular VLP array"

Response 6: Changed references to VLP arrays to cellular VLP array

Point 7: I wonder if Figure 4C is a western blot analysis or is it rather SDS-PAGE analysis with total protein staining?

I would suggest to present both SDS-PAGE with total protein staining and western blot analysis.

Response 7: Figure 4C is a western blot utilizing antibody F1412SA.

Reviewer 3 Report

The authors present a novel platform to produce FMDV VLPs that can be used to vaccinate cattle and pigs against disease. The described method will allow the production of VLPs at facilities that do not have high containment facilities, thus enabling many countries where FMD is endemic to manufacture these with no risk. The study demonstrates the potential usefulness of these VLPs as vaccines in both swine and cattle. However, future work will be needed to determine the efficacy as this was only a proof of principle study. Nevertheless, the authors were successful in the demonstration that this new platform can be usefully used on other FMDV serotypes. I will be looking forward to see the results of the clinical trial for this vaccine platform in cattle especially.

Major comment:

None

Minor comments:

Line 113: Add TEM abbreviation here. First use.

Line 199: Only use abbreviation here as already introduced TEM in Line 113.

Lines 201-202: Remove components. Already mentioned in Materials and Methods.

Fig 3: Appropriately label the Y-axis.

Fig 4B: Add arrow that highlights the 1.38g/cm3 layer.

Line 257: Superscript 3.

Line 279: Should an be and?

Author Response

Point 1: Line 113: Add TEM abbreviation here. First use.

Response 1: Added

Point 2: Line 199: Only use abbreviation here as already introduced TEM in Line 113.

Response 2: Changed

Point 3: Lines 201-202: Remove components. Already mentioned in Materials and Methods.

Response 3: Removed

Point 4: Fig 3: Appropriately label the Y-axis.

Response 4: Added

Point 5: Fig 4B: Add arrow that highlights the 1.38g/cm3 layer.

Response 5: Added in the text description of the figure that the 1.38g/cm3 layer is between the two marks on the tube and added a blue box behind the tubes to better emphasize the layer.

Point 6: Line 257: Superscript 3.

Response 6: Changed the 3 in all C3I references to C3I

Point 7: Line 279: Should an be and?

Response 7: Added a comma after swine to reduce confusion

Round 2

Reviewer 2 Report

I would like to thank the authors for thorough response. Before acceptance, I would strongly recommend to include the SDS-PAGE analysis of the (semi)purified VLPs into the supplementary material. Also, it would be important to add a statement into the manuscript text indicating that the VLP preparates were not free of contaminating proteins.

Author Response

Point 1: Before acceptance, I would strongly recommend to include the SDS-PAGE analysis of the (semi)purified VLPs into the supplementary material. Also, it would be important to add a statement into the manuscript text indicating that the VLP preparates were not free of contaminating proteins.

Response: Added the Coomassie blot into the supplemental materials and the following statement into materials and methods section 2.4 "This extraction methodology produces vaccine preparate containing proteins from the production cell line in addition to VLPs.  A Coomassie stained SDS-PAGE gel of an O1M vaccine preparate is presented in supplementary material."  

The authors have also added a statement in the acknowledgements thanking Ms. Janine Simmons for her assistance in running the SDS-PAGE gel and Coomassie staining.